# Towards the elimination of FGM by 2030: A statistical assessment

**Kathrin Weny** *, **Romesh Silva, Rachel Snow, Berhanu Legesse, Nafissatou Diop**

Technical Division, United Nations Population Fund, New York, New York, United States of America

* weny@unfpa.org

## Abstract

In 2015, UN member states committed to eliminate female genital mutilation (FGM) by 2030 as part of the Sustainable Development Agenda. To reach this goal, interventions need to be targeted and guided by the best available evidence. To date, however, estimates of the number of girls and women affected by FGM and their trends over time and geographic space have been limited by the availability, specificity and quality of population-level data. We present new estimates based on all publicly available nationally representative surveys collected since the 1990s that contain both information on FGM status and on the age at which FGM occurred. Using survival analysis, we generate estimates of FGM risk by single year of age for all countries with available data, and for rural and urban areas separately. The likelihood of experiencing FGM has decreased at the global level, but progress has been starkly uneven between countries. The available data indicate no progress in reducing FGM risk in Gambia, Guinea-Bissau, Mali and Guinea. In addition, rural and urban areas have diverged over the last two decades, with FGM declining more rapidly in urban areas. We describe limitations in the availability and quality of data on FGM occurrence and age-at-FGM. Based on current trends, the SDG goal of eliminating FGM by 2030 is out of reach, and the pace at which the practice is being abandoned would need to accelerate to eliminate FGM by 2030. The heterogeneity in trends between countries and rural vs urban areas offers an opportunity to contrast countries where FGM is in rapid decline and explore potential policy lessons and programmatic implications for countries where the practice of FGM appears to remain entrenched.

## Introduction and background

Female genital mutilation (FGM) refers to all procedures involving the partial or total removal of the external female genitalia or other injury to the female genital organs for non-medical reasons [1]. As early as 1994, the Programme of Action of the International Conference on Population and Development (ICPD), called on governments to eliminate FGM [2]. Elimination of FGM requires targeted programming of proven interventions, the identification of which can only be determined by tracking reliable estimates of age-specific incidence through operational or observational trials.

The global population map of girls at risk of FGM is evolving over time, as select communities and countries abandon the practice, while others report little change. Locating those at

**Data Availability Statement:** All data is taken from the publicly available Demographic and Health Surveys (http://dhsprogram.com) and Multiple Indicator Cluster Surveys (http://mics.unicef.org).

**Funding:** The author(s) received no specific funding for this work.

**Competing interests:** The authors have declared that no competing interests exist.

greatest risk of FGM requires a careful review of global and national age-specific trends over the last few decades, and likely future trajectories. As we mark the 25-year anniversary of the ICPD, it is timely to review these estimates in FGM incidence in the context of the 2030 Sustainable Development Agenda, which includes a global goal to eliminate FGM by 2030.

UNICEF has estimated that at least 200 million women worldwide live with FGM [3]. This number is useful for raising awareness within the international community and to inform the public about the scale of the problem. However, prevalence estimates are not optimal for policy makers and program specialists trying to develop evidence-based actions and plan effectively for the future, as estimates of FGM prevalence include women who experienced the practice decades earlier. Periodic estimates of age-specific risks for FGM in a given country provide a better basis by which to assess underlying secular trends, and hence, the impact of various policy and/or programmatic interventions to end FGM. Age-specific risk for FGM, in this regard, is defined as the probability of a girl to experience FGM at a certain age, if she has not experienced FGM previously in her life.

Prevailing estimates of FGM suffer from a variety of shortfalls. An estimate that roughly 3.3 million girls are at risk of experiencing FGM per year is based on applying prevalence rates among adolescents age 15–19 directly to populations of girls 0–14 years [4]. Yet, where age-specific data are available, the FGM risk for 15-19-year olds is frequently not the same as that for girls age 0–14 years (S1–S6 Figs). In general, selecting broad global age categories to estimate FGM prevalence, and using such estimates to compare FGM risk between diverse countries is imprecise due to stark differences in the age-specific risks of cutting between countries. Furthermore, population prevalence estimates based on FGM among girls younger than age 15 can under-estimate the risk of FGM [5], as girls may still be cut at later ages. The precision of such an approach is limited by the fact that girls in the age group 0–14 include some who have already experienced FGM, some who have not yet experienced FGM but will in future, and others who will never experience FGM. Age-at-FGM is very heterogeneous across countries, and hence national age-specific risks of FGM must be accounted for within any aggregate estimate of global levels and trends. This will also affect to trend estimates, as changes in age-at-FGM over time would be translated into changes in prevalence.

Estimating the magnitude and trend in FGM incidence poses significant challenges, given the paucity and quality of FGM data, the complexity of both survey design and data structure. Maximizing the effective use of available data requires accommodating the different age-specific risks of FGM for girls and women between different countries, combining these with the latest and best population projections for each country, accounting for incomplete information, and recognizing that girls may be cut at later ages.

We present age-specific estimates of FGM risk for birth cohorts from the 1960s until today, across 24 countries. To our knowledge, this is the first multi-country study to model and estimate the age-specific risk for FGM by incorporating all publicly available data with standardized FGM information since the early 1990s, including past and present FGM risk.

## Materials and methods

The data sources for this analysis are all Demographic and Health Surveys (DHS) and Multiple Indicator Cluster Surveys (MICS) with a module on FGM including information on status and age-at-FGM. These data have been pooled into a dataset of 67 nationally representative household surveys from 24 countries. Despite more than 90 DHS and MICS surveys currently being available that contain an FGM module, our pooled analysis excludes subnational areas that were not continuously included in the sampling frame of national DHS and MICS surveys, surveys that were not nationally representative, those that only included ever-married women,

and surveys that did not contain age-at-FGM (S1 Appendix and S1 Table). Our inclusion criteria for MICS and DHS surveys are principled and appropriate for the synthetic cohort modeling approach used, while also maintaining geographic comparability as well as survey coverage depending on marital status. Other studies include more surveys and cover more countries, some of which do not contain FGM survey modules with age-of-FGM data or are not comparable across successive surveys for a given country [5].

Questions on FGM have been included in Demographic and Health Surveys since the 1990s [6], but the availability of nationally representative data on FGM risk and age-at-FGM varies considerably across countries. Only a few countries, such as Senegal, Tanzania and Mali, have routinely included FGM data within the DHS over the last three decades. Intervals between surveys vary widely between countries, and some countries have only a single survey that included data on FGM (e.g. Cameroon and Yemen). This uneven data availability leads to challenges when attempting to reliably estimate trends at global, regional, national, and subnational levels.

During the last three decades, the period during which these survey data were collected, the countries in our analysis have experienced substantial socio-economic and political changes including shifting administrative boundaries, as well as changes in the security and accessibility of certain regions. The humanitarian crises in Mali, for example, prevented 2012–13 DHS survey field teams from accessing three regions in their entirety, and a fourth region partially [7]. As the excluded regions had been included in both previous and later DHS surveys, this survey is not directly comparable to others at the national level. We accounted for these inconsistencies by excluding data from regions that had not been continuously included in survey coverage, or excluding surveys that were not nationally representative (S1 Appendix).

An additional limitation of the available data is that prior to 2010 DHS and MICS survey questions on FGM suffered from substantial time lag between the time of the FGM event and the time of reporting. Before the sixth DHS wave, the women's questionnaire FGM module addressed only respondents aged 15–49 and asked, among others, about their FGM status, the age at which they suffered FGM, which form of FGM they experienced, and who performed it. However, it did not include information on all their living daughters aged 0–14 and only incorporated a standardized question on the total number of daughters that experienced FGM. It subsequently inquired the age-at-FGM and other details about the daughter that was most recently cut [8]. This information does not allow us to reconstruct an FGM-risk-profile of all daughters. Consequently, even for the youngest cohort among respondents, i.e. women aged 15–19 years, FGM may have occurred more than a decade before data collection, making it extremely hard to reliably assess recent trends.

Around 2010 both DHS and MICS surveys introduced a daughter's module that includes questions on FGM occurrence and age-at-FGM for each respondent's daughter aged 0–14 years. Data collected via the daughters' module has reduced the time lag between reporting and the time of FGM events. However, a daughter reported as not having experienced FGM at the time of the survey may still be affected by it in future; thus, the FGM module contains right-censored data.

We combined both the mother's and daughter's modules from available DHS and MICS surveys. These include the standard FGM module on each women age 15–49, with questions on her knowledge about the practice, whether she has experienced FGM herself, when it happened and who performed it. This module also includes the type of FGM she experienced and, often, if the woman thinks the practice should continue or not [9]. In the daughter's module, each woman age 15–49 years is asked to provide information on behalf of their daughter(s) aged 0–14 years about her FGM status, current age, age-at-FGM, FGM type, and person who performed FGM. We have used the 'survey' package version 3.36 from Thomas Lumley to perform Kaplan-Meier estimates in complex survey design [10].

To study the global trend of FGM occurrence over time, we applied survival analysis and grouped women and girls in all countries into one-, five- and 10-year birth cohorts. In order to implement time-to-event analysis, where the event of interest is FGM occurrence, we re-ordered the data as shown in Fig 1.

Our measure of interest is the probability of not experiencing FGM before age, *t*, represented by the survival function *S(t)*:

$$S(t) = P(T > t)$$

where *T* is the time at which the girl or woman experiences FGM

By using the number of 'events' (i. e. cases of FGM) in each period (i.e. year of life of the full cohort of women and girls) and the number of girls at risk, we estimate *S(t)* from birth to age 49 using the Kaplan-Meier method [11, 12]. Our estimates are restricted to the age range from 0 to 49 years due to limitations of the DHS and MICS survey data.

$$\hat{S}(t) = \prod_{t=0}^{49}\left(1 - \frac{d_t}{n_t}\right)$$

where *d* = number of events, i. e. FGM cases,

*n* = girls and women at risk.

*t* = time in age of life.

To account for the complex survey designs used in DHS and MICS surveys, which entail multi-stage cluster sampling, we applied a Horvitz-Thompson approach to obtain unbiased variance estimates [11] based on the work of Swensson [13]. This takes into account the

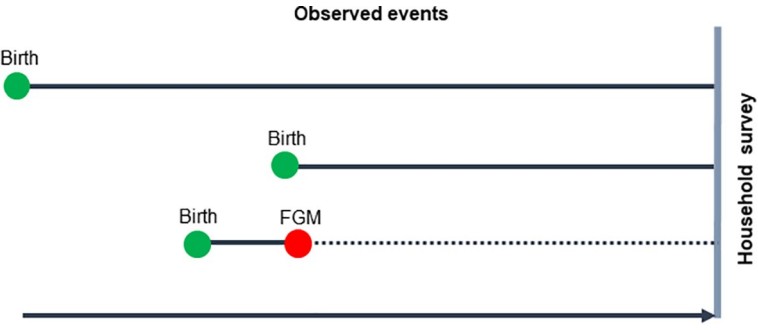

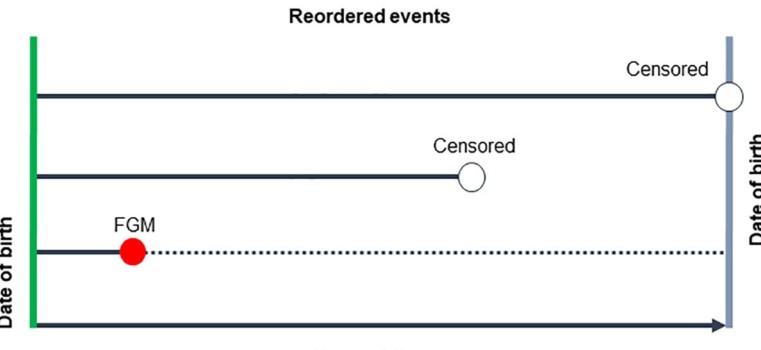

**Fig 1. Survival analysis with the FGM module.**

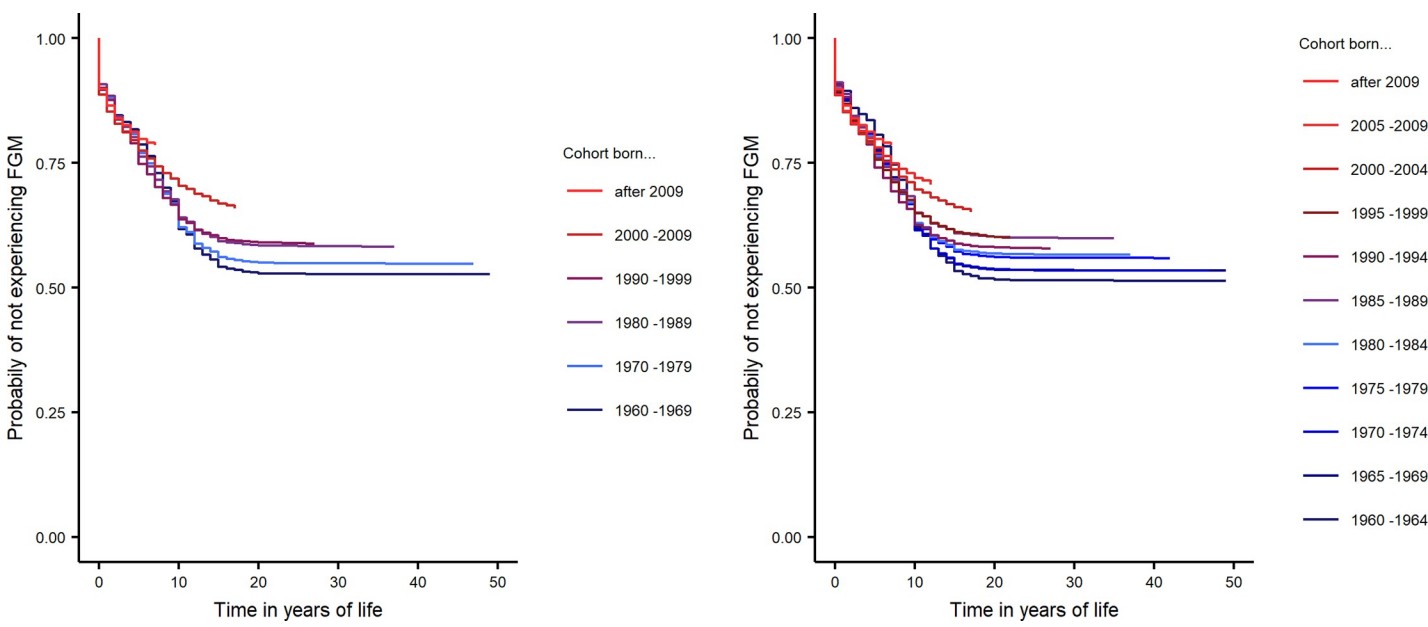

**Fig 2.** (A) Survival curves for 10-year age cohorts show decline in FGM risk from the 2000s. (B) Survival curves for 5-year age cohorts show decline in FGM risk from the late 1990s.

probability of a girl, i.e. her mother, to be included in the last stage of the survey, given the survey design in the preceding stages [13]. We used the individual sampling weights that account for differences in sampling probability due to the specific survey design and response rates [14, 15]. We do not adjust sampling probabilities when using data from the FGM daughter's module. As data on all daughters aged 0–14 years of women aged 15–49 years are included in the surveys, no further adjustment of sampling weights is required [14]. For the pooled analysis, sampling weights were de-normalized using population estimates from the latest World Population Prospects [16].

## Results

Our descriptive Kaplan Meier analysis demonstrates a substantial decline in the risk of FGM throughout the last six decades (Fig 2A and Table 1). The survival curves for each successive cohort are stacked on top of one another, with older cohorts, born in 1960–1969 or 1960–1964 respectively, on the bottom and younger cohorts above. The analysis based on five-year birth cohorts (Fig 2B, and Table 2) shows that this decline in FGM risk accelerated in the late 2000s.

**Table 1. Unweighted count of women and girls per 10-year cohort.**

| Cohort | Unweighted count of women and girls |
| --- | --- |
| 1960–1969 | 83,940 |
| 1970–1979 | 176,103 |
| 1980–1989 | 248,912 |
| 1990–1999 | 229,819 |
| 2000–2009 | 274,744 |
| After 2009 | 139,390 |

**Table 2. Unweighted count of women and girls per 5-year cohort.**

| Cohort | Unweighted count of women and girls |
|---|---|
| 1960–1964 | 27,647 |
| 1965–1969 | 56,293 |
| 1970–1974 | 78,856 |
| 1975–1979 | 97,247 |
| 1980–1984 | 120,601 |
| 1985–1989 | 128,311 |
| 1990–1994 | 127,175 |
| 1995–1999 | 102,644 |
| 2000–2004 | 122,108 |
| 2005–2009 | 152,636 |
| After 2009 | 139,390 |

## Women and daughters living in rural vs urban households

In order to localize the decline in FGM and to determine areas that have not experienced a drop of FGM risk, the following section presents descriptive statistics by rural and urban residence of the respondent across all countries represented in our dataset.

In our analysis, we observe the onset of the FGM decline with the age cohorts born in the 2000s for both rural and urban areas (dark red line in Fig 3). However, soon afterwards, we recognize a faster pace of decline in FGM risk for urban areas than rural ones.

Fig 3 further demonstrates that women and girls in rural areas have been exposed to significantly greater risk of FGM than those living in urban areas across all age cohorts. From the 1960s to the mid-1980s, the shape of the survival curves also indicates that rural/urban differences in FGM-risk were due to less cutting at later ages (10+) in urban areas, as the survival curves overlap until year 10–12. This spread in FGM-risk in later years of life may be also due

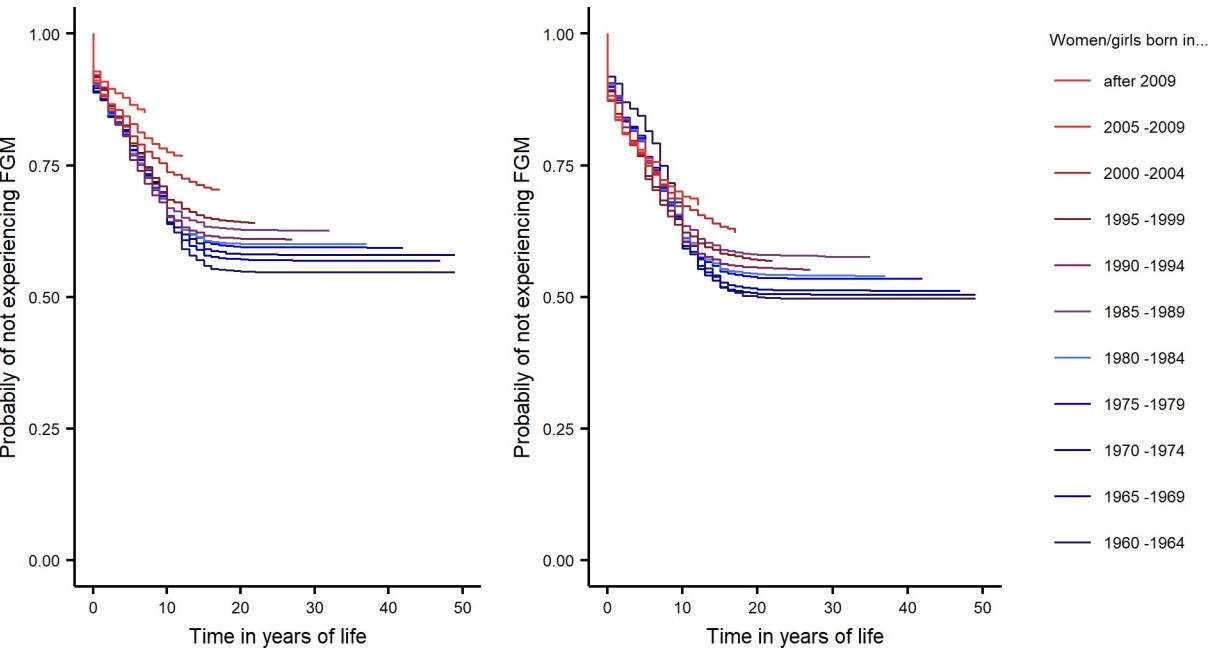

**Fig 3. Faster decline in FGM is observed in urban compared to rural areas.** (A) Urban areas. (B) Rural areas.

to age misreporting and that girls and women in urban areas are generally subject to a lower risk of FGM across all ages.

From 1990 onwards, this relationship changes, with girls in urban areas having less risk of FGM in each year of life, and the urban/rural difference in FGM risk slowly increases across all subsequent five-year cohorts.

This progressive distinction between rural/urban FGM risk for cohorts born after 1990 coincides with the time in which we observe an acceleration in aggregate decline in FGM at the global level, shown in Fig 4, and is evident within the national pattern of decline for countries such as Guinea, Burkina Faso, Central African Republic and Cote d'Ivoire (S1–S6 Figs). This suggests that much of the progress in eliminating FGM has occurred in urban areas.

The difference in the probability of experiencing FGM between rural and urban areas is highly significant across the period under review, highlighting the geographic clustering of declines in FGM. The effect of rural versus urban residence may be attributed to its correlation to women's status [17], and exposure to media [18], or paternal and maternal education [19].

## National results

It is important to note that the aggregations as displayed above depend on data availability across countries for different cohorts. For example, all cohorts born after the year 2000 no longer contain data on Yemen as the last survey in the country was conducted in 2013 and did not include questions on the daughters of the survey respondents. This warrants a disaggregation and analysis of trends and levels of FGM at national level. Our analysis indeed demonstrates substantial variation of FGM decline across the 24 countries included in our analysis (S1–S6 Figs). Particularly in the case of Guinea, Guinea-Bissau, Gambia, Mali, and Cameroon, we find no evidence of a decline of FGM risk, while all other countries experienced significant reductions.

Fig 5A and 5B highlight the diversity in FGM risk profiles between countries, by displaying survival curves for all cohorts in three countries with substantial decline in FGM (Ethiopia, Mauritania and Kenya) and three countries with little evidence of a decline in the practice (Gambia, Guinea and Guinea-Bissau). These survival curves display differences in FGM with respect to 1) the overall, or lifetime, risk of experiencing FGM for women and girls demonstrated by the height of the survival curves, 2) the existence and pace of FGM decline at the national level, portrayed by the relative height of survival curves for different cohorts and 3) the age-pattern of the practice depicted by the shape of the survival curves and how it was influenced by FGM decline.

Women and girls in Guinea experience the highest lifetime risk of FGM compared to women and girls from the same age cohorts in all other countries. This is indicated by the lowest point of the survival curves in the figure for Guinea, which is reached at about age 20. In contrast, Kenya or Guinea-Bissau, particularly, display a lower lifetime risk of FGM.

In addition, the Kaplan-Meier estimates demonstrate little change in FGM risk over six decades in Guinea, Gambia and Guinea-Bissau as we see overlapping survival curves. In Kenya, Mauritania and Ethiopia the survival curves are stacked on top of one another beginning in the 1960s, indicating a consistent decline in FGM risk for young cohorts of women/girls throughout the period of observation. In the case of Ethiopia, we observe large jumps in the survival curves from one cohort to the next, particularly for girls born in the late 1990s to today. In Mauritania, especially the two youngest cohorts, born in 2005–2009 and born after 2009, seem to experience less risk of FGM, while in Kenya a consistent and steady decline seems to have been achieved throughout the six decades covered in our analysis.

Moreover, the figures allow us to analyze at what particular age women and girls are most at risk of experiencing FGM and if this risk profile has changed in the last decades. In the case

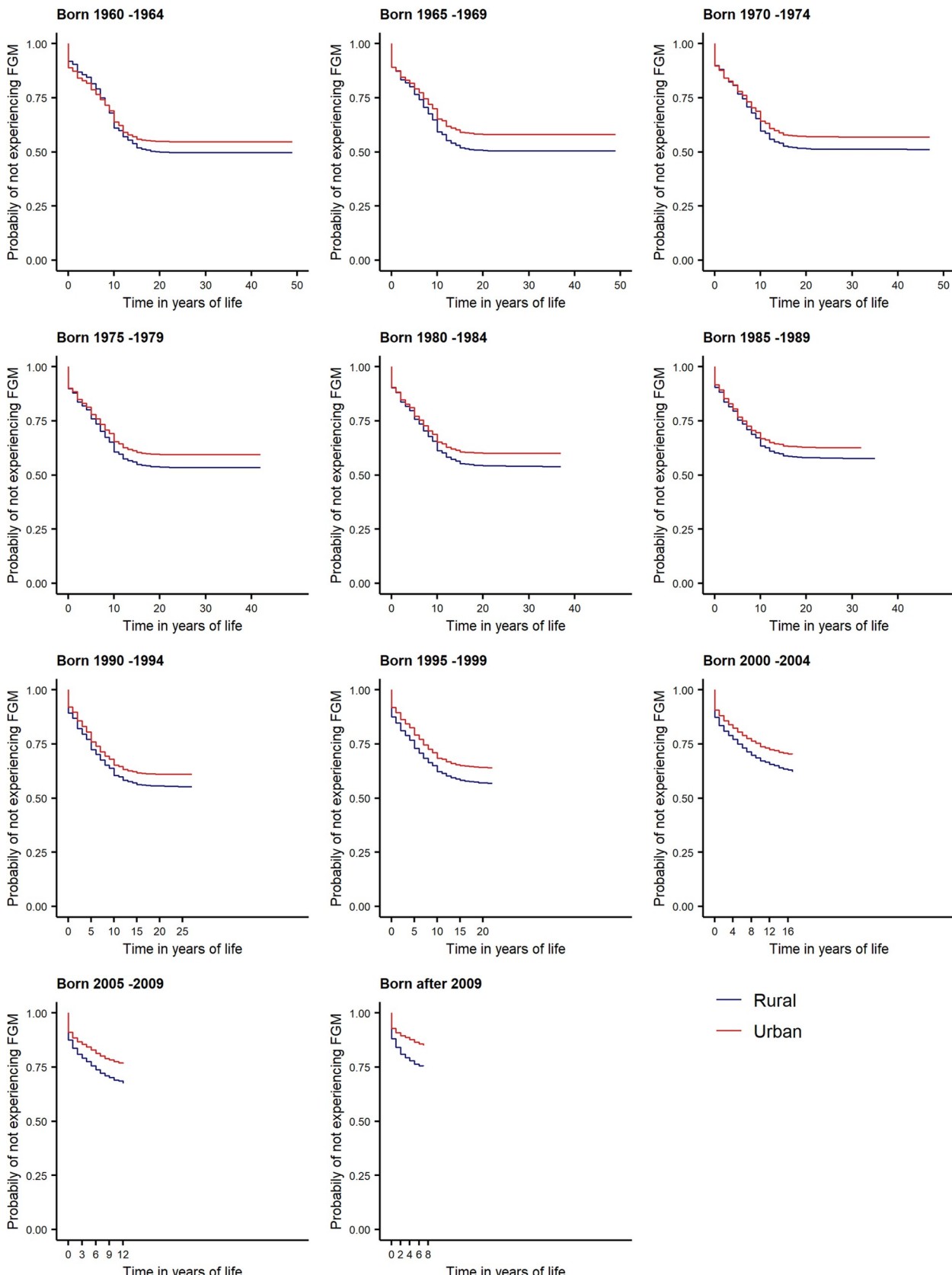

**Fig 4. FGM dynamics by 5-year cohorts show increasing disparity in FGM risk between rural and urban areas.**

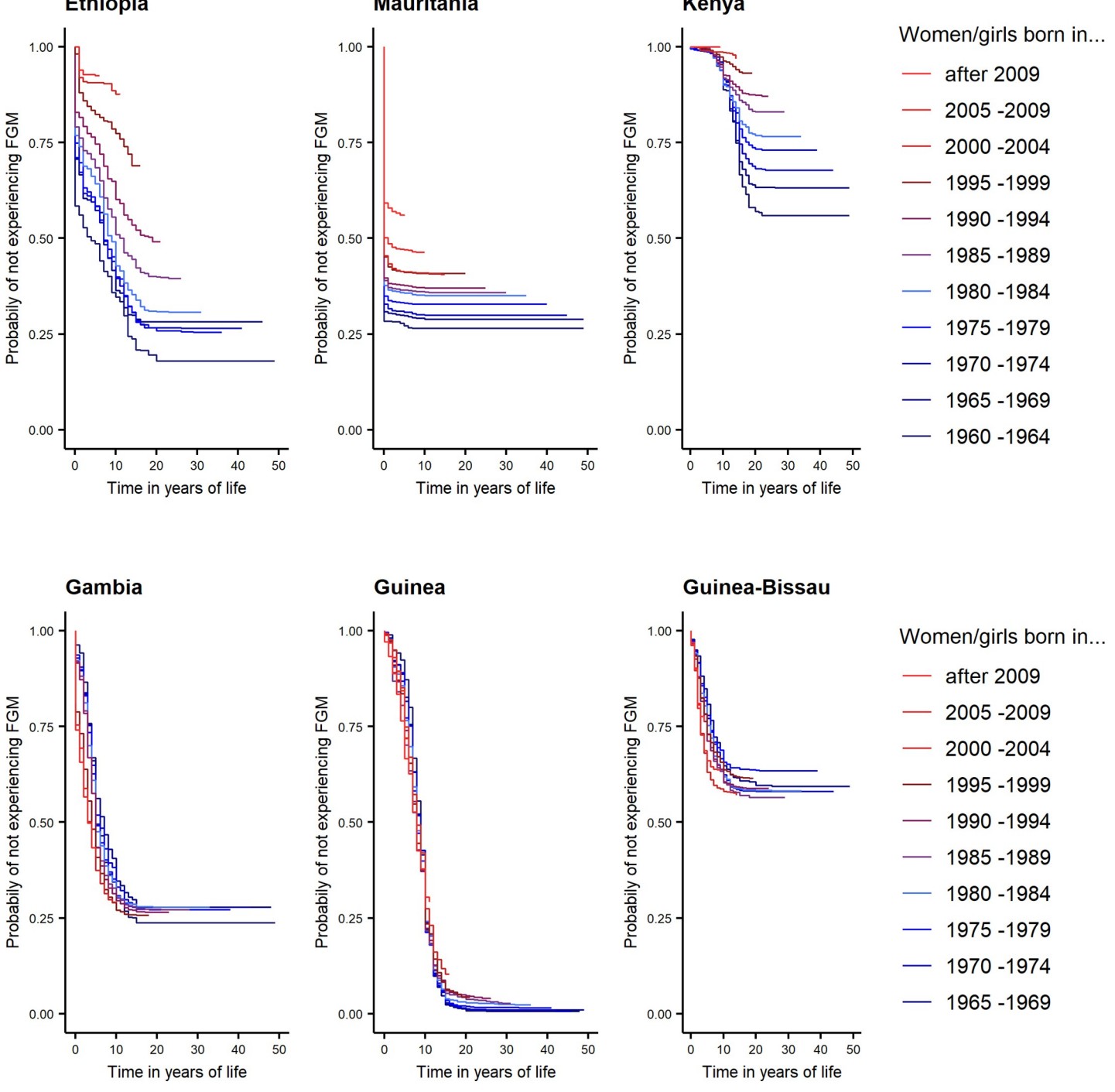

**Fig 5.** (A) Decline in FGM risk in Ethiopia, Mauritania and Kenya. (B) Lack of decline in FGM risk in Gambia, Guinea and Guinea-Bissau.

of Ethiopia and particularly Kenya girls are at risk of FGM until their late teens, while in Mauritania, most of the FGM risk takes place in the very first years of a girl's life. This has important implications for programming. Where FGM is carried out on infants, interventions targeted at parents, birth attendants, health facilities and other institutions are vital, while cutting at later ages allows interventions through girl empowerment, alternative rites of passages

and interventions in primary and secondary schools. Finally, the age at FGM, stayed relatively constant across these countries as the shape of the survival curves remain almost unchanged for all age cohorts. This finding is consistent with the substantive literature classifying FGM as a social norm [20, 21].

## Discussion

To our knowledge, this is the first study that uses all publicly available survey micro datasets with FGM occurrence and age-at-FGM data to retrospectively estimate age-specific FGM risk for different age cohorts per single year of age. These new estimates exploit the direct and indirect reporting on FGM risk in household surveys, adjust for age-specific risk within a flexible framework of survival analysis, and do not assume prevalence of FGM risk and age-at-risk are constant. We make use of the timelier reporting of FGM occurrence in DHS and MICS surveys, by fully incorporating proxy reporting by women aged 15–49 about their daughters aged 0–14 years, via the FGM daughters' questionnaire module. While individual DHS and MICS surveys have indicated a decline in FGM prevalence [5, 22–26], to date there has been no standardized, multi-country estimate of the overall decline over the last few decades, illustrating the timing of the onset, and comparing the pace of decline between different countries.

In this study, we present estimates of FGM risk by birth cohorts for 24 countries with available data from 1960 to today, presenting Kaplan-Meier survival curves to estimate relative annual decline in FGM risk by residence and at the national level. We note that the recent major declines in FGM risk are heavily concentrated in urban areas. This suggests that further reductions in FGM risk will only be possible if program interventions can successfully target and reach subpopulations of girls in hard-to-reach, remote communities—and raises additional questions on the nature, feasibility and scale of FGM prevention programs needed for further reductions.

Further, our analysis shows that there is substantial heterogeneity between countries in the overall risk of FGM, age-at-FGM and pace of decline. On the one hand, we find no decline for Gambia, Cameroon, Mali, Guinea or Guinea-Bissau, while countries such as Ethiopia or Mauritania show strong signs of decline, and even an acceleration of the downward trend in recent decades. The different FGM risk patterns demonstrated in our analysis provide programming intervention experts with more precision on when girls are most at risk.

In short, the estimates and analysis presented herein point to areas of commendable progress, to areas where more attention and improved interventions are needed, and where and when these interventions are best conducted–to advance FGM elimination by 2030. It also points towards countries where no progress has been made and where the elimination of FGM by 2030 is not feasible if the country stays on its current trajectory.

Given the fact that our dataset combines both mother- and daughter-specific exposure data and our outcome variable measures an event that occurred possibly decades before the survey data were collected, potential socio-demographic covariates of interest do not apply consistently to the person-at-risk of FGM. For example, socio-demographic characteristics such as education are collected for the survey respondent only, i.e. the woman age 15–49.

We are therefore limited to using household characteristics as socio-economic controls. However, even these suffer from the time lag between the time of survey data collection and the event occurrence. We cannot track if, for example, the residence of the household has changed and at the moment FGM occurred a recently urbanized household still lived in a rural area, or if a now wealthy household used to live in poverty. In addition, the wealth index is derived separately from each survey, limiting its comparability across surveys and making a direct comparison of socio-economic status of households problematic [27].

Our national level survival analyses are subject to data gaps, data quality issues and modeling/estimation challenges. In particular, some countries such as Senegal have multiple surveys in recent years that include the needed information on occurrence and timing of FGM, whereas other countries have one DHS or MICS survey (e.g. Yemen) or no publicly available data, particularly micro datasets (e.g. Indonesia). Without increased availability of FGM survey microdata from more countries, a comprehensive assessment of FGM trends, and hence the needed targeted interventions, is not possible.

To make this feasible, data collection beyond DHS and MICS data is indispensable. DHS and MICS suffer from limitations such as the lack of lower level geographic disaggregation which makes it impossible to identify and target small-scale hotspot areas. Baseline data for programmatic intervention are one option and allow modelling FGM decline at smaller geographic levels and statistical progress assessment in areas covered by programmatic interventions. Routine data collection in the health sector is another way to enlarge the variety of data sources that current FGM estimates are based on.

Further, more validation studies are needed to account for the nature and quality of self-reporting and proxy-reporting of FGM risk in household surveys. Indirect interviewing techniques have revealed substantial biases in responses to FGM questions [28]. Validation analyses are particularly important in understanding how to coherently combine and model self-reported risk by adult women and proxy reporting by mothers about their daughters, and the reliability of these data as recall bias increases with age and as national legal frameworks and community social norms concerning FGM shift over time.

Finally, in order to obtain truly global level and trend estimates of FGM risk, including for countries where FGM has been reported anecdotally [29], more representative surveys are needed, including in countries that are traditionally not covered by DHS and MICS surveys or subnational communities that may be best served by other measurement approaches. Research on FGM practices in Europe [30, 31], the United States [32], and Asia and the Pacific [33] have been emerging, but more work has to be done to standardize these methods [34]. Such assessment, enhancement and standardization are fundamental for an evidence-based pathway to zero FGM by 2030.

In 2003, Toubia and Sharief wrote: "The greatest obstacle to a scientific answer [of a decline in FGM] is the fact that reliable baseline data were not collected (. . .)" [35]. More than 15 years later, we are still confronted with this problem, but we have attempted to address this by constructing a baseline synthetic cohort and estimating decline through cohorts thereafter.

## Supporting information

**S1 Fig. Country-level results: Benin, Cameroon, Burkina Faso and Central African Republic.**
(TIF)

**S2 Fig. Country-level results: Chad, Egypt, Côte d'Ivoire, Ethiopia.**
(TIF)

**S3 Fig. Country-level results: Gambia, Guinea, Ghana, Guinea-Bissau.**
(TIF)

**S4 Fig. Country-level results: Iraq, Mali, Kenya, Mauritania.**
(TIF)

**S5 Fig. Country-level results: Niger, Senegal, Nigeria, Sierra Leone.**
(TIF)

**S6 Fig. Country-level results: Sudan, United Republic of Tanzania, Togo, Yemen.**
(TIF)

**S1 Appendix. Data exclusions.**
(DOCX)

**S1 Table. Data sources and percentage of "During infancy" replies.**
(DOCX)

## Author Contributions

**Conceptualization:** Kathrin Weny, Rachel Snow, Nafissatou Diop.

**Formal analysis:** Kathrin Weny.

**Methodology:** Kathrin Weny, Romesh Silva.

**Project administration:** Berhanu Legesse.

**Supervision:** Rachel Snow, Nafissatou Diop.

**Visualization:** Kathrin Weny.

**Writing – original draft:** Kathrin Weny, Romesh Silva.

**Writing – review & editing:** Rachel Snow, Berhanu Legesse.

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
