## [Decision Letter · Decision Letter 0]

18 May 2020

PONE-D-19-32389

Eliminating FGM - is the 2030 agenda target within reach?

PLOS ONE

Dear Mrs. Weny,

Thank you for submitting your manuscript to PLOS ONE. After careful consideration, we feel that it has merit but does not fully meet PLOS ONE’s publication criteria as it currently stands. Therefore, we invite you to submit a revised version of the manuscript that addresses the points raised during the review process.

This is a very timely paper and we ask that you please address the minor points made by the reviewers to add clarity to your paper.

We would appreciate receiving your revised manuscript by Jul 02 2020 11:59PM. To enhance the reproducibility of your results, we recommend that if applicable you deposit your laboratory protocols in protocols.io, where a protocol can be assigned its own identifier (DOI) such that it can be cited independently in the future. For instructions see: http://journals.plos.org/plosone/s/submission-guidelines#loc-laboratory-protocols

We look forward to receiving your revised manuscript.

Kind regards,

Fiona Cuthill, PhD

Academic Editor

PLOS ONE

Journal Requirements:

2. Please modify the title to ensure that it is meeting PLOS’ guidelines (https://journals.plos.org/plosone/s/submission-guidelines#loc-title).

In particular, the title should be "specific, descriptive, concise, and comprehensible to readers outside the field" and in this case it is not informative and specific about your study's scope and methodology.

Please ensure that you amend both the title on the online submission form (via Edit Submission) and the title in the manuscript so that they are identical.

3. Please amend your manuscript to include your abstract after the title page.

4. We note you have included tables to which you do not refer in the text of your manuscript.

Please ensure that you refer to Tables 1 and 2 in your text; if accepted, production will need this reference to link the reader to the Table.

Reviewers' comments:

Reviewer's Responses to Questions

**Comments to the Author**

1. Is the manuscript technically sound, and do the data support the conclusions?

Reviewer #1: Yes

Reviewer #2: Yes

2. Has the statistical analysis been performed appropriately and rigorously? 

Reviewer #1: Yes

Reviewer #2: Yes

3. Have the authors made all data underlying the findings in their manuscript fully available?

Reviewer #1: Yes

Reviewer #2: Yes

4. Is the manuscript presented in an intelligible fashion and written in standard English?

Reviewer #1: Yes

Reviewer #2: Yes

5. Review Comments to the Author

Reviewer #1: Review Comments: Eliminating FGM - is the 2030 agenda target within reach?: PONE-D-19-32389

The authors report a secondary quantitative study titled “Eliminating FGM - is the 2030 agenda target within reach? Applying survival analysis to analyze FGM risks on a cross sectional dataset from 24 countries.

The paper adopts a novel methodological approach to present critical findings and suggestions that can inform FGM-related interventions. The methodological approach and suggestions are bold leveraging on rich DHS/MICS FGM data collected over a long period to confidently inform policy and programming interventions. This paper is well thought and reads very well. Nonetheless I have some comments as follows;

Methods

1. There is need to define risk of FGM? In this case a statement on that DHS/MICS data show the current FGM status of the girl/women as well as the risk for FGM should be included. This is important not to confuse the reader.

2. Page 4 line 70 “Our inclusion criteria for MICS and DHS surveys are principled and appropriate for the synthetic cohort modeling approach we use. Other studies include more surveys and cover more countries, some of which do not contain FGM survey modules with age-of-FGM data or are not comparable across successive surveys for a given country. It would be helpful to the readers to include some of the major inclusion and exclusion criteria.

3. It will be helpful to describe how many DHS/MICS data sets were included that were for before 2010 as well as those that were obtained after 2010?

4. Since the DHS/MICS FGM module has a question on the type of FGM, it would have been helpful to analyses the data around this component to see the age specific risk to help us understand role of health awareness campaigns in FGM.

5. The DHS/MICS FGM module has a question on who performed FGM. It would have been helpful to analyses the data around this component to see the age specific risk to help us infer the trends towards medicalization

6. An underdeveloped piece in this paper is the role of adoption and enactment of anti-FGM legislations in various countries around 2000s and later. This could have an impact on the DHS/MICS data because of underreporting.

Discussion

1. I would be interested to see a concluding statement/paragraph in view of the current analysis and the projection for elimination of FGM by 2030.

2. Page 15 line 295-300: “In 2003, Toubia and Sharief wrote: “The greatest obstacle to a scientific answer [of a decline in FGM] is the fact that reliable baseline data were not collected (…)” [34]. More than 15 years later, we are still confronted with this problem, but we have attempted to address this by constructing a baseline synthetic cohort and estimating decline through cohorts thereafter”. What suggestions do the author have on improving FGM data collection, processing and presentation?. Are they of the view that that DHS/MICS could be inadequate? Or what improvements need to be done.

Reviewer #2: This is an important and timely study of FGM/C to understand the extent which has been achieved in reducing this practice globally. Although the available data is rather limited in some countries, but the statistical analysis is rigorous and the result is supporting the main argument. This study provides clear trends of the decline of FGM/C risk in some countries which useful for strategic intervention planning.

Hitherto, the discussion of FGM/C is predominantly focus on Africa and Middle East. There is a paucity of data from Asia Pacific region. For a brief picture of FGM/C in this region please refer to Dawson, A., Rashid, A., Shuib, R., Wickramage, K., Budiharsana, M., Hidayana, I. M., & Marranci, G. (2020). Addressing female genital mutilation in the Asia Pacific: the neglected sustainable development target. Australian and New Zealand Journal of Public Health.

One minor correction, Indonesia has a publicly data on FGM/C, please check https://www.litbang.kemkes.go.id/laporan-riset-kesehatan-dasar-riskesdas/

6. PLOS authors have the option to publish the peer review history of their article (what does this mean?). If published, this will include your full peer review and any attached files.

Reviewer #1: Yes: Dr. Samuel Kimani

Reviewer #2: No

---

## [Author Response · Author response to Decision Letter 0]

2 Jul 2020

Dear Sir or Madam,

We would like to thank PlosOne’s academic editor and the peer reviewers for their helpful comments and suggestions with respect to our manuscript as well as providing us with the opportunity to submit a revision.

The style requirements, including file names, positioning of abstracts as well as title of the manuscript have been amended according to the style requirements of PlosOne. The updated version of our manuscript has been uploaded in the submission form, as well as in the ‘Revised Manuscript with Track Changes’.

Below we have drafted responses to all comments and questions by the reviewers listing amendments we have made within the manuscript. 

Reviewer 1:

I. On the definition of risk of FGM

We have included a particular note in the introduction outlining the definition of ‘risk of FGM’ as one of experiencing FGM at a given age IF one has not experienced FGM earlier in life. This provides a more explicit definition compared with the previous version. This note can be found in the ‘Introduction and background’ section. 

II. On the inclusion criteria of DHS/MICS surveys

With respect to this question, we would like to note that ‘S7- Appendix. Data exclusions describes surveys that are not part of the analysis including the exclusion criteria. As per recommendations by the reviewers, we have also added the main exclusion criteria to the main body of the text in the ‘Materials and methods’ section. 

III. On the distinction between DHS/MICS before 2010 and after

S8 Table in the Annex describes each single survey including survey year and number of responses that are ‘during infancy’. The cut-off between surveys that included a module on daughters cannot be strictly made in the year 2010, as countries, such as Iraq, do not include a daughter module, even after 2010. Therefore, the full list of countries in S8 - Table is necessary for complete transparency and has now been added in the appendix.

IV. On the inclusion of analysis of the type of FGM

The type of FGM is indeed included in the vast majority of FGM survey modules both for DHS and MICS. However, we have not included an analysis of type of FGM in this analysis and estimated the overall risk of FGM of all types for several reasons.

First and foremost, entrenched gender inequality is behind all types of FGM, independent of their invasiveness. In addition, all forms of FGM are associated with increased health risk and violates the sexual and reproductive health and rights of women and girls.

With the adoption of the Sustainable Development Goals, the UN and its member states have pledged to eliminate the practice by 2030, in all types and forms, and our analysis is providing a background evaluation of the feasibility of this pledge. Zero tolerance to all forms of female genital mutilation is the guiding principle of this analysis.

V. On the inclusion of analysis regarding medicalization of FGM

It is true that medicalization of female genital mutilation (practice of female genital mutilation by health care providers) is an alarming trend which requires due attention. Medicalization of female genital mutilation is more common in certain countries and programmes need to respond to this context in these countries. However, discussion on medicalization is beyond the scope of this article. For the article, the main emphasis is to project the number of girls at risk of female genital mutilation based on population data instead of going into details of who performs the practice. 

VI. On the inclusion of analysis on anti-FGM legislation

This is a very valuable comment with respect to reliability of survey results retrieved from self-reporting by women, and proxy reporting of women with respect to their daughters.

We have included that validation analysis is of particular importance given the recall bias as well as the changing landscape of national legal frameworks in paragraph 9 of the discussion section. 

We feel that given the other potential sources of biases, such as the self-reporting for women and the proxy reporting for girls, the different age groups and the extent of the recall bias, among others, the point is covered with due justice and further elaboration is beyond the scope of the current analysis.

VII. On the inclusion of more language connecting the current analysis and the projection for elimination of FGM by 2030

This is a very helpful comment, and we have added a further explanation in paragraphs 4 and 10 of the discussion section on the meaning of the results of our analysis for the probability of the global elimination of FGM by 2030.

VIII. On the question on improving FGM data collection, processing and presentation

We have added more details with respect to increased data collection efforts beyond DHS/MICS that would be necessary to estimate FGM trends (Please see paragraphs 6 and 9 of discussion section). However, it is important to emphasize that there has been improvement in the questionnaire used by DHS/MICS (such as having a module to report on the prevalence for 10-14 years of old) and trend analysis is becoming possible for different countries as there are different data sets from DHS/MICS and through cohort approaches.

We have added further possible expansion of quantitative data sources for the estimation of FGM in the ‘Discussion’ section, such as the increased use of administrative records. 

Reviewer 2:

IX. On the inclusion of data form the Asia and Pacific region, and in particular Indonesia

We agree with the inclusion of this academic resource to enhance our discussion section (paragraph 6 of discussion section). In this section, we maintain that in order that countries in which FGM has been reported anecdotally have to be included in ‘truly global level and trend estimates’, and the suggested literature review on FGM in Asia and the Pacific is a meaningful addition (paragraph 9 of discussion section).

Indeed, Indonesia has conducted a national survey in 2013 including information on the FGM for girls age 0-11. The current analysis, however, required micro data in order to be able to perform the estimations.

We amended the relevant reference in the text to make clear that the lack of data in the case of Indonesia refers to the lack of publicly available micro datasets, not published prevalence estimates in general (paragraph 6 of discussion section).

With kind regards, 

Kathrin Weny, Romesh Silva, Rachel Snow, Berhanu Legesse, and Nafissatou Diop

---

## [Decision Letter · Decision Letter 1]

25 Aug 2020

Towards the elimination of FGM by 2030: A statistical assessment

PONE-D-19-32389R1

Dear Dr. Weny,

We’re pleased to inform you that your manuscript has been judged scientifically suitable for publication and will be formally accepted for publication once it meets all outstanding technical requirements.

Kind regards,

Fiona Cuthill, PhD

Academic Editor

PLOS ONE

Additional Editor Comments (optional):

Reviewers' comments:

Reviewer's Responses to Questions

**Comments to the Author**

1. If the authors have adequately addressed your comments raised in a previous round of review and you feel that this manuscript is now acceptable for publication, you may indicate that here to bypass the “Comments to the Author” section, enter your conflict of interest statement in the “Confidential to Editor” section, and submit your "Accept" recommendation.

Reviewer #1: All comments have been addressed

Reviewer #2: All comments have been addressed

2. Is the manuscript technically sound, and do the data support the conclusions?

Reviewer #1: Yes

Reviewer #2: Yes

3. Has the statistical analysis been performed appropriately and rigorously? 

Reviewer #1: Yes

Reviewer #2: Yes

4. Have the authors made all data underlying the findings in their manuscript fully available?

Reviewer #1: Yes

Reviewer #2: Yes

5. Is the manuscript presented in an intelligible fashion and written in standard English?

Reviewer #1: Yes

Reviewer #2: Yes

6. Review Comments to the Author

Reviewer #1: Review Comments: Towards the elimination of FGM by 2030: A statistical assessment: PONE-D-19-32389R1

The authors report a secondary quantitative study titled “Towards the elimination of FGM by 2030: A statistical assessment? Applying survival analysis to analyze FGM risks on a cross sectional dataset from 24 countries.

The paper adopts a novel methodological approach to present critical findings and suggestions that can inform FGM-related interventions. The methodological approach and suggestions are bold leveraging on rich DHS/MICS FGM data collected over a long period to confidently inform policy and programming interventions.

The manuscript has substantially and significantly improved, while the authors have comprehensively addressed the comments that were raised earlier.

I am satisfied with the level of improvement of the manuscript and responses to the comments by the authors.

Reviewer #2: (No Response)

7. PLOS authors have the option to publish the peer review history of their article (what does this mean?). If published, this will include your full peer review and any attached files.

Reviewer #1: **Yes: **Samuel Kimani

Reviewer #2: No

---

## [Editor Report · Acceptance letter]

3 Sep 2020

PONE-D-19-32389R1 

Towards the elimination of FGM by 2030: A statistical assessment 

Dear Dr. Weny:

I'm pleased to inform you that your manuscript has been deemed suitable for publication in PLOS ONE. Congratulations! Your manuscript is now with our production department. 

Kind regards, 

on behalf of

Dr. Fiona Cuthill 

Academic Editor

PLOS ONE